# Prototyping and Evaluation of Graphene-Based Piezoresistive Sensors

**Lucas Florêncio, Jéssica Luzardo, Marcelo Pojucan, Victor Cunha, Alexander Silva** **, Rogério Valaski and Joyce Araujo ***

Divisão de Metrologia de Materiais (Dimat), Instituto Nacional de Metrologia, Qualidade e Tecnologia (Inmetro), Av. Nossa Senhora das Graças, 50, Duque de Caxias 25250-020, Brazil; lflorencio@inmetro.gov.br (L.F.); jmluzardo@inmetro.gov.br (J.L.); mpsantos-prometro@inmetro.gov.br (M.P.); vrcunha@colaborador.inmetro.gov.br (V.C.); amartins@inmetro.gov.br (A.S.); rvalaski@inmetro.gov.br (R.V.)

* Correspondence: jraraujo@inmetro.gov.br

**Abstract:** In this work, the electrical properties of graphene papers were investigated with the aim of developing pressure sensor prototypes for measuring pressures up to 2 kPa. In order to determine which graphene paper would be the most suitable, three different types of graphene papers, synthesized by different routes, were prepared and electrically characterized. The results of electrical characterizations, in terms of electrical conductivity and sheet resistance of graphene papers, are presented and discussed. Prototypes of pressure sensors are proposed, using graphene papers obtained by chemical oxidation (graphene oxide and reduced graphene oxide) and by electrochemical exfoliation. The prototypes were tested in static compression/decompression tests in the working range of 0 kPa to 1.998 kPa. The compression/decompression sensitivity values observed in these prototype sensors ranged from 20.8% ΔR/kPa for graphene sensors obtained by electrochemical exfoliation to 110.7% ΔR/kPa for those prepared from graphene oxide obtained by chemical oxidation. More expressive sensitivity values were observed for the sensors fabricated from GO, intermediate values for those made of rGO, while prototypes made of EG showed lower sensitivity.

**Keywords:** graphene paper; piezoresistivity; prototype; pressure sensor

## 1. Introduction

Graphene, first obtained experimentally in 2004, exhibits a series of exceptional properties which, since its discovery, have stimulated several studies for the development of electronic devices [1]. In addition to its remarkable electronic properties, graphene also has extraordinary mechanical properties. The combination of these outstanding mechanical and electronic properties qualifies graphene as an ideal material for application in production of electromechanical sensors [2,3].

Graphene can be manufactured in a paper form, with thickness in the order of hundreds of micrometers (<200 μm), by stacking graphene monolayers [4]. With this structural condition, graphene-based sensing devices can be prototyped in reduced dimensions compared to traditional ones [5]. This material, in micrometric films, has excellent mechanical properties with a large deformation coefficient which enables high sensitivity when used in pressure sensing devices. Its small size and high sensitivity make it promising for applications such as pressure, acoustic and mass sensors in the biomedical, environmental, microsystems and nanosystems areas [6]. Additionally, these sensors have stable and reversible responses, such as being able to be made flexible and transparent, in addition to having an adjustable working range [7–10].

The development of graphene-based pressure sensors is promising due to their exceptional electromechanical properties [11]. For this application, its piezoresistive property is noteworthy: its resistivity varies linearly with dimensional deformation when subjected to external mechanical compression loads, allowing the sensors to operate linearly over a wide

pressure range [12]. This piezoresistive effect has been widely used in the field of smart sensors, including pressure sensors [13]. Recent studies about graphene-based pressure sensors include wearable piezoresistive physical sensors [14], highly sensitive, reliable and flexible piezoresistive pressure sensors [15], sensors for human health monitoring [16], flexible, highly sensitive and wearable pressure and strain sensors with graphene porous network structures [17], and ultralightweight and 3D squeezable graphene–polydimethylsiloxane composite foams as piezoresistive sensors [18].

Graphene-based pressure sensors can exhibit high sensitivity, responding with electrical signals to pressure variations. The sensor presented by Chun et al. achieved a sensitivity response of 17% $\Delta R/10$ kPa in a working range from 1 kPa to 100 kPa [7]. For the sensor presented by Kazemzadeh et al., the response was $-19$ mV kPa$^{-1}$ in a working range of 1 kPa to 10 kPa [8]; the graphene-based sensor pressure developed by Habibi et al. showed a response of 21.9 ($\mu$A/A)/Pa in a working range of 100 kPa to 170 kPa [9].

The values recorded for detection limits and response time of these sensors are also remarkable. The sensor presented by Vaka et al. [10] recorded values for response time smaller than 15 ms and detection limit smaller than 0.6 Pa. Considering the flexible and transparent sensor structure, such properties make it suitable for various applications such as robotics, smart screens, artificial skin and health monitoring.

The main characteristics of graphene-based pressure sensors, as a function of their type of construction/response, are shown in Table 1. As can be seen in the table, the sensors presented different responses when subjected to pressure variations.

**Table 1.** Characteristics of graphene-based pressure sensors.

| Sensor | Response [1] | Sensitivity | Working Range | Reference |
|---|---|---|---|---|
| Piezoresistive (flexion) | $\uparrow P = \uparrow R$ | $17 \pm 7\%$ $\Delta R/10$ kPa | (1–100) kPa | [7] |
| Piezoresistive (compression) | $\uparrow P = \downarrow R$ | $-19$ mV/kPa | (1–10) kPa | [10] |
| Piezoresistive (dual: flexion and compression) [2] | $\uparrow P = \uparrow R$ and $\uparrow P = \downarrow R$ | 4.2 $\Delta R$/kPa and 29.4 $\Delta R$/kPa | (0.01–10) kPa | [7] |
| Field emitter | $\uparrow P = \uparrow I$ | 21.9 ($\mu$A/A)/Pa | (0.01–10) kPa | [8] |
| Diaphragm | $\uparrow P = \uparrow \delta$ | 36 and 63 nm/kPa | (0–5) kPa | [6] |

[1] Variables evaluated: *P* (pressure), *R* (electrical resistance), *I* (electric current), *δ* (mechanical deformation); [2] the behavior will change depending on the applied pressure range.

This work presents the fabrication of prototype graphene-based pressure sensors, including their electrical and mechanical characterization, and evaluation of their performance in the detection of mechanical compression forces. Two graphene papers were obtained by chemical synthesis methods, graphene oxide (GO), reduced graphene oxide (rGO) and a third one by electrochemical synthesis, named here electrochemically synthesized graphene (EG). The prototype sensors fabricated with these graphene papers were tested and compared in terms of their piezoresistive responses when subjected to controlled compression forces.

## 2. Materials and Methods

Graphene Papers

Graphene is the basic material for the construction of the proposed sensor prototypes. The graphene materials used were graphene oxide (GO), reduced graphene oxide (rGO) and electrochemically synthesized graphene (EG). All graphene papers used in the construction of the sensor and in the preparation of the samples were processed in the

Surface Phenomena Laboratory, Lafes, of the National Institute of Metrology, Quality and Technology (INMETRO).

GO synthesis was performed with expanded graphite purchased from Nacional de Grafite Ltda (São Paulo, Brazil). Sodium nitrate ($NaNO_3$), potassium permanganate ($KMnO_4$), sulfuric acid ($H_2SO_4$), hydrochloric acid (HCl) and hydrogen peroxide ($H_2O_2$) were purchased from Sigma-Aldrich (St. Louis, MO, USA). Calcium carbonate ($CaCO_3$, 98.5%) was purchased from Merck (Darmstadt, Germany). EG synthesis was performed using a 0.25-mm-thick graphite foil obtained from Alfa Aesar, and ultrapure water (Milli-Q water, Millipore system) with resistivity $\geq$ 18 M$\Omega$ cm$^2$. Solutions were made from N, N'-dimethylformamide (99%) obtained from Sigma-Aldrich.

The chemical route used in this work to obtain GO papers was based on the oxidation of graphite powder in the presence of acids and strong oxidants in accordance with the Hummers method [19]. In order to obtain the reduced graphene oxide, rGO, by removing the oxygenated functional groups, the GO received a thermal treatment in an inert atmosphere (Ar) allowing for a partial restoration of the carbon framework which brought this graphene closer to its original physical structure [20]. Similarly, EG was synthesized according to the method proposed by Parvez et al. [21] through electrochemical exfoliation of a graphite foil, used here as a working electrode, and a wire platinum as a counter-electrode in a $(NH_4)_2SO_4$ electrolyte solution containing intercalating chemical agents. When an electrical tension of 10 V is applied in this electrochemical cell for a duration of 30 min, a reduction reaction occurs and considerably improves the graphene's electrical and electronic properties [21,22].

The graphene papers were produced using a vacuum filtration system for the graphene material dispersed in dimethylformamide (DMF). Graphene papers made with graphene oxide produced by electrochemistry (EG) and graphene oxide graphene papers produced by chemical route (GO) were filtered. After the end of the filtration, the graphene papers were taken into an oven for complete drying of the solvent. The GO paper, alternatively, was treated in an oven with an inert atmosphere to reduce the oxygen functional groups, originating the rGO paper.

Morphological Characterization of the Graphene Papers

Scanning Electron Microscopy (SEM) of the fabricated graphene papers was performed using a Field Emission Scanning Electron Microscope (FEG-SEM-FIB), model Helios NanoLab 650 (FEI-Thermo Fischer Scientific, Hillsboro, OR, USA), operated at 1 kV, 25 pA.

Electrical Characterization of the Graphene Papers

Considering that the graphene-derived graphene papers have a piezoresistive behavior, the first step of the membrane characterization was the estimation of the sheet resistance of different graphene papers through I-V curves under DC conditions, which allowed for comparison of the electrical behavior of the three types of membrane with different electrical characteristics: graphene oxide (GO), reduced graphene oxide (rGO) and electrochemically exfoliated graphene (EG). GO is an insulating material, while rGO and EG are conductive with EG exhibiting a higher conductivity in comparison to rGO [7]. In order to obtain the DC I-V curves, square graphene papers (10 × 10 mm) with Ag contacts were prepared. The metal contacts (100 nm) were produced through thermal resistive evaporation under high vacuum conditions ($10^{-7}$ Torr) in an Angstrom Engineering Vacuum Chamber.

For the I-V curve determination, a Keithley Source Meter 2400 voltage source was used with needle-type probes as electrodes to evaluate current variation as a function of applied voltage for each type of membrane. The voltage range was −2 V to 2 V with a controlled step of 0.02V. For each sample, 12 measurements were carried out in order to evaluate material stability.

Plots I-V show an ohmic behavior, which allowed for the estimation of sheet resistance through linear fitting.

Fabrication of the Prototypes

In order to evaluate the membrane response to mechanical load variations, the graphene papers made from EG, rGO and GO were cut into squares of with sides of 19 mm. Copper sheets (thickness of 0.1 mm and 10 mm × 30 mm in dimension) were positioned on the upper and lower surfaces of the membrane. The membrane and copper sheet assembly were then wrapped in an adhesive vinyl insulator 19 mm wide and 0.12 mm thick. The ends of the copper sheets were exposed for electrical coupling. The sensor prototype is illustrated in Figure 1a.

Electrical–Mechanical Tests

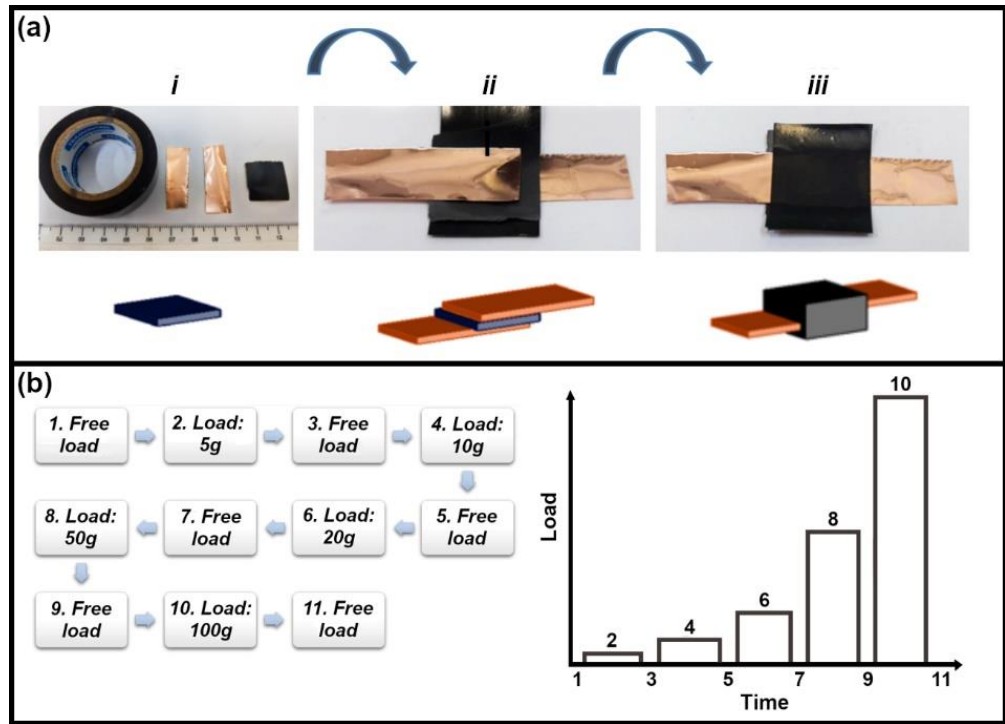

**Figure 1.** Preparation of the sensor prototype where: (**a**) membrane cut (i); copper foil assembly (ii) and insulator assembly (iii); (**b**) scheme illustrating loading and unloading cycles with the values of the applied weights and illustration of computed data, load x time.

The experiment proposed to evaluate the sensor prototype was the static mechanical test. In this test, the sensor prototype is laid onto a test bench and its terminals are connected to Keithley Source Meter 2400 voltage source probes. In this setup, the sensor prototype was supported by a smooth, flat surface. A constant voltage was applied to the sample while current variation under the compression of different loads was recorded. The loads were standard masses of 5 g, 10 g, 20 g, 50 g and 100 g, and the first measurement was recorded without compression. After mounting the sensor on the measurement bench and connecting it to the voltage source, the electrical current was evaluated from the condition of no-load to maximum load in the sequence (cycle) as shown in Figure 1b. The applied mechanical load (standard mass block) was converted into pressure, considering contact area (between the mass and the sensor prototype) and the respective weight force of each mass used (Figure 1b). For each sensor prototype, the following parameters were evaluated as a function of the applied mechanical load:

1   Variation of electrical current (and the corresponding variation of electrical resistance);
2   Definition of the characteristic value of average electrical resistance for each applied mechanical load;
3   Sensitivity of the sensor prototype, in terms of electrical resistance variation by applied load, (ΔR)/kPa.

Seven (7) sensor prototypes were tested: three (3) made of GO, three (3) of EG and one (1) of rGO. The rGO paper presented a fragile and brittle behavior, which made it hard to prepare other prototypes of this material besides the first one. Thus, only one specimen of rGO was evaluated in this study.

## 3. Results and Discussion

### 3.1. Morphological Characterization

The surface morphologies of the EG and GO papers were analyzed by SEM. The SEM image of the EG sheet, Figure 2A, shows a crumpled texture typical of exfoliated graphene sheets [23]. By contrast, the SEM image of GO paper, Figure 2C, shows a smooth surface. Additionally, Figure 2B,D show the cross-section stacking of EG and GO papers. The stacking height of EG paper is higher than GO paper. In these images, one can see typical wrinkles of graphene and some higher regions (120 nm for both GO and EG), indicating that both papers have regions with different numbers of graphene layers stacked on top of each other. EG is known to have significant heterogeneity and may be more oxidized/exfoliated in some parts than others, presenting some non-oxidized/non-exfoliated regions such as the original graphite [23]. Topography heterogeneity as well as different types of defects (edge defects, vacancies, functional groups, heteroatoms) in graphene sheets, mainly generated by the electrochemical oxidation and exfoliation processes, may generate electrical mobility, explaining the higher electrical conductivity of EG papers when compared to GO papers, as observed in I-V curves, Figure 3.

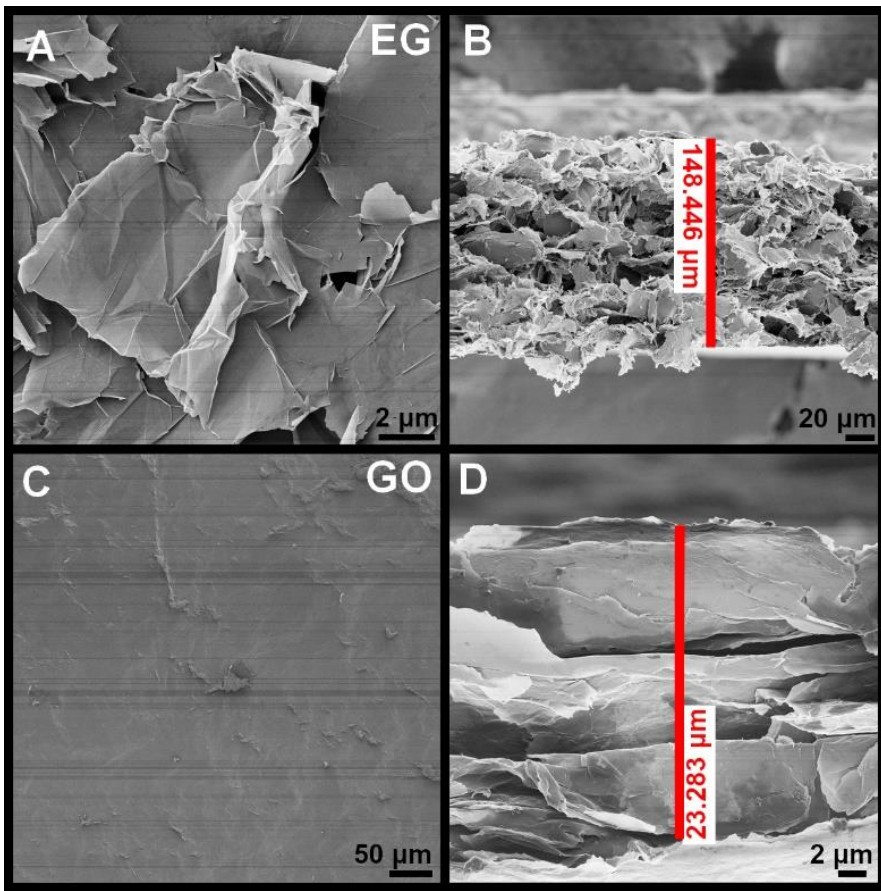

**Figure 2.** SEM images of the (**A**) surface and (**B**) cross-section of the EG paper, and (**C**) surface and (**D**) cross-section of the GO paper.

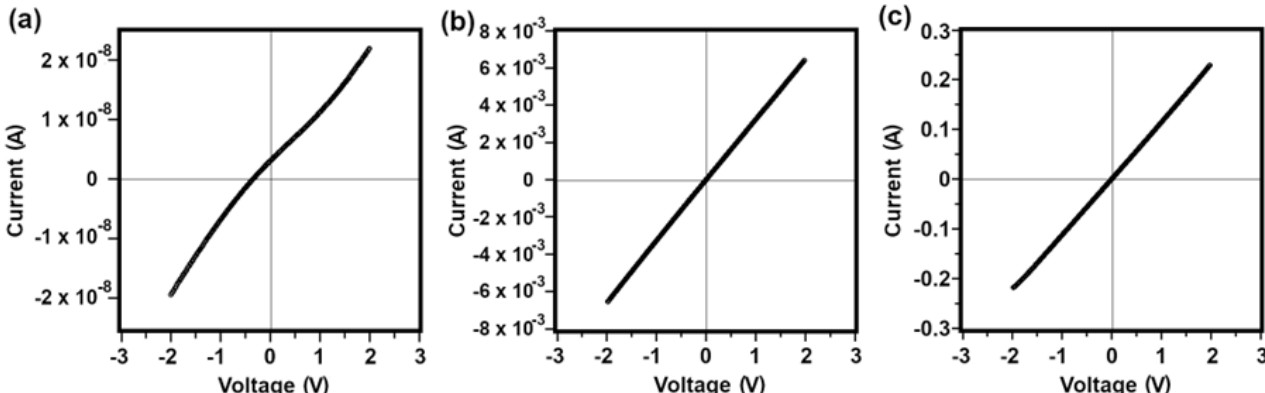

**Figure 3.** Voltage (V) × current (A) plots for the graphene samples: (**a**) GO1, (**b**) rGO1 and (**c**) EG.

### 3.2. Electrical Characterization

Five graphene samples were electrically characterized: two GO papers, two rGO papers and one EG paper. Figure 3 presents the current versus voltage graphs, I-V, for three representative graphene papers, one of each synthesis method. The current and voltage data, referring to the electrical measurements of these graphene papers, indicate that the current is proportional to the electrical voltage applied to the rGO and EG samples, as shown by the linearity of the plotted curve. For the GO paper, this behavior was not totally linear in the region close to the origin (in the region of small values of voltage and current in these tests). The curve measured for GO was displaced from the origin in the positive direction, indicating the presence of a residual current (positive) at zero voltage.

The sheet resistance values found in these measurements, shown in Figure 3, were: $7.44 \times 10^7$ Ω for GO1, $3.03 \times 10^2$ Ω for rGO1 and 8.81 Ω for EG. Such values indicate that the GO papers have much higher electrical resistance than the EG papers (seven orders of magnitude), while rGO papers have intermediate values between those of EG and GO, which is consistent with the literature [24].

### 3.3. Mechanical Test

In the proposed test, the variation in electrical resistance as a function of the variation in mechanical load was evaluated, showing the capacity of the prototype to respond to controlled variations in mechanical loads from the zero-load condition to the maximum applied load of 1.998 kPa. Increase in the value of mechanical load on the prototype leads to a decrease in electrical resistance. This behavior can be attributed to the increase in contact area between graphene layers in the membrane under compression, which in turn promotes a higher electrical conductivity due to the proximity of the charge carriers.

As expected, for those prototypes built from EG, the absolute variation in electric current values is more expressive than those observed in the GO-based prototypes. For rGO, the values are intermediate. This fact is explained by the electrical properties of EG papers, having good electrical conductivity, while GO has an insulating character and rGO has an intermediate behavior. The absolute values of electrical current variation and their respective conversion into electrical resistance, calculated for the GO prototype, revealed high values for electrical resistance, of the order of $10^9$ Ω while the electrical current is in the order of $10^{-8}$ Å. As shown in Figure 4, the electrical response for GO exhibited a considerable amount of noise, which may have been caused by the fact that the current is close to the detection limit of the equipment. For the rGO and EG tests, the curves are smoother. The values of electrical current variation are in the order of $10^{-1}$ Å for the EG prototype and $10^{-2}$ Å for the rGO prototype. Variations in electrical resistance of 10 Ω and 100 Ω for EG and rGO, respectively, were computed.

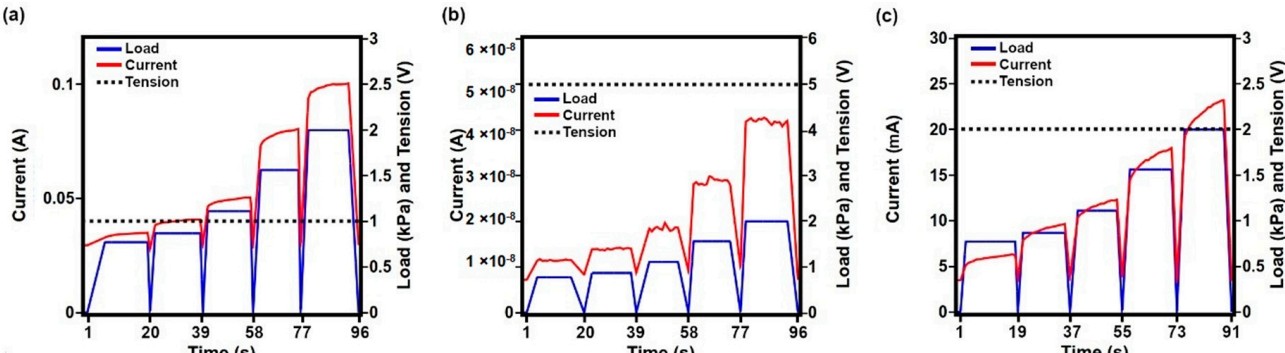

**Figure 4.** Graphs of static electromechanical tests: current × pressure × voltage plots for different graphene papers; (**a**) EG1, (**b**) GO1 and (**c**) rGO1.

Figure 4 illustrates the measurement cycles where it is possible to observe that sensor response is extremely fast, stable under load application and removal, and reversible, i.e., when loading and unloading are applied to the prototype, the electric current increases which maintains this value under load, and when it is unloaded, the current returns to its characteristic value of zero load.

Figure 5 shows sensitivity (S) versus applied load. Sensitivity can be defined in terms of resistance variation per kilopascal as well as relative sensitivity, which considers the specific resistance variation, for each tested prototype in terms of percentage of resistance variation per kilopascal. For EG prototypes, absolute sensitivity is in the order of 10 $\Omega$/kPa, $10^2$ $\Omega$/kPa for rGO and $10^9$ G$\Omega$/kPa for GO.

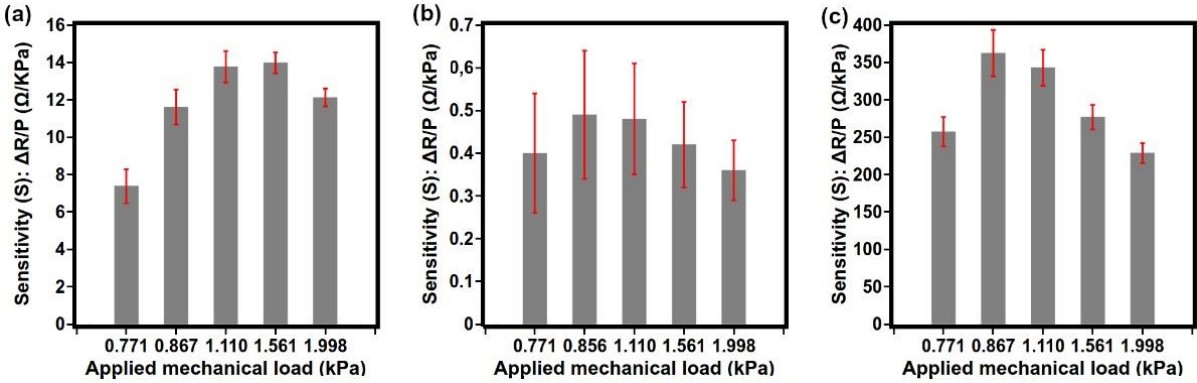

**Figure 5.** Sensitivity (S) for each load applied to the sensor prototypes; (**a**) EG1, (**b**) GO1 and (**c**) rGO1.

In terms of relative sensitivity, the GO-based sensor prototypes showed higher sensitivity in comparison to rGO and EG ones, with an average of up to 110.7% $(\Omega/\Omega)$/kPa for the mechanical load of 0.771 kPa, as shown in Table 2. The sensitivity values found for the GE-based prototypes ranged from 20.8% $(\Omega/\Omega)$/kPa to 69.48% $(\Omega/\Omega)$/kPa. Finally, the rGO1 prototype tested showed relative sensitivity values with mean values ranging from 42.2% to 66.8%.

Thus, the GO-based prototypes, a material that has the characteristic of an electrical insulator, presented higher absolute values of electrical resistance and the largest relative variations of these resistances when subjected to variations in mechanical load. Despite having lower conductivity values, these GO-based prototypes recorded the highest sensitivity values when compared to EG- and rGO-based prototypes.

The proposed sensor prototype has a piezoresistive response to compression in which an increase in pressure causes a decrease in resistance, observed in the test through the increase in electrical current. Such a response occurs because the compression load promotes an approximation between the graphene sheets (and possible functionalized particles)

which leads to an increase in contact area between the sheets and, consequently, to an increase in electrical conductivity [7,8,10].

**Table 2.** Relative sensitivity, *s*, of sensor prototypes and their respective standard deviations [% (Ω/Ω)/kPa]; (a) EG1, (b) EG2, (c) EG3, (d) GO1, (e) GO2, (f) GO3, (g) rGO1.

| Sensor Prototype | $s_{0.771}$ [(Ω/Ω)/kPa] | $s_{0.867}$ [(Ω/Ω)/kPa] | $s_{1.110}$ [(Ω/Ω)/kPa] | $s_{1.561}$ [(Ω/Ω)/kPa] | $s_{1.998}$ [(Ω/Ω)/kPa] |
|---|---|---|---|---|---|
| EG1 | −20.8%; 2.5% | −32.8%; 2.3% | −38.9%; 1.5% | −39.5%; 0.8% | −34.2%; 0.7% |
| EG2 | −38.8%; 6.2% | −53.4%; 6.3% | −52.6%; 3.4% | −43.8%; 2.2% | −38.7%; 1.5% |
| G3 | −65.68%; 4.96% | −69.48%; 4.85% | −68.30%; 1.96% | −53.94%; 0.87% | −43.97%; 0.61% |
| GO1 | −45.71%; 10.57% | −56.10%; 8.99% | −55.52%; 6.09% | −48.69%; 3.37% | −42.13%; 1.33% |
| GO2 | −77.3%; 12.9% | −90.0%; 6.3% | −76.2%; 3.3% | −60.2%; 0.6% | −48.5%; 0.3% |
| GO3 | −110.66%; 2.40% | −102.33%; 1.71% | −82.16%; 0.81% | −60.25%; 0.57% | −48.41%; 0.20% |
| rGO1 | −47.4%; 1.8% | −66.8%; 2.7% | −63.2%; 1.3% | −51.1%; 0.5% | −42.2%; 0.3% |

It was noticed that the prototype sensors respond to small pressure variations, and the sensitivity was evaluated for each applied load. Compared to other piezoresistive graphene sensors, the prototypes showed promising sensitivities [7,10], ranging from 20.8% ΔR/kPa up to 110.7% ΔR/kPa.

The responses obtained by the sensor prototypes in the working range used in this evaluation provide evidence regarding the piezoresistive behavior of graphene papers. Absolute and relative sensitivities were calculated from the analysis of variations in electrical properties. The working range was from 0 kPa to 1.998 kPa. This limited range of work observed here, however, may be extended in future works. It is important to consider that the results presented by the prototypes in the present work do not allow us to qualify their response as linear, as it was noticed that the sensitivity of these prototypes depends on the applied mechanical load.

**4. Conclusions**

In this work, the electrical properties of graphene papers were investigated with the aim of developing pressure sensor prototypes. A mechanical test model was established to evaluate the prototypes when subjected to differential compression forces. In the electrical characterization step, the proposed test allowed us to evaluate the sheet resistance of the graphene papers and to know their electrical behavior. The EG material exhibited high electrical conductivity, while GO had an insulating behavior and rGO an intermediate behavior. With the data obtained for conductivity as a function of each type of graphene paper, a sensor prototype was proposed based on a simple assembly model. The pressure sensor prototypes were evaluated in static mechanical tests in which mechanical loads, composed of a set of standard masses, exerted compression forces on the sensors. This test allowed for the response of the prototypes to variations in mechanical load to be evaluated within the proposed working range of 0 kPa to 2 kPa by measuring the characteristic resistance for each applied load. This test allowed us to demonstrate sensitivity values for each tested prototype, also making it possible to compare results between prototypes. More expressive sensitivity values were observed for sensors fabricated from GO, intermediate values for those made of rGO, while prototypes made of EG showed lower sensitivity.

**Author Contributions:** Conceptualization, L.F. and J.A.; methodology, L.F., J.L. and J.A.; software, V.C.; investigation, L.F.; resources, J.L., M.P., R.V.; data curation, L.F.; writing—original draft preparation, L.F.; writing—review and editing, A.S. and J.A.; visualization, L.F. and A.S.; supervision, J.L. and J.A.; project administration, J.A.; funding acquisition, J.A. All authors have read and agreed to the published version of the manuscript.

**Funding:** The authors would like to thank Faperj (Foundation for Research Support of the State of Rio de Janeiro) and CNPq (*Conselho Nacional de Desenvolvimento Científico e Tecnológico*) for fellowship support: grants E-26/202.746/2018 and 311900/2017-8, respectively. We would like to thank Braulio Soares Archanjo for the Scanning Electron Microscopy image acquisition.

**Conflicts of Interest:** The authors declare no conflict of interest.

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
