# Peer review of "Prototyping and Evaluation of Graphene-Based Piezoresistive Sensors"

_electronicmat, doi:10.3390/electronicmat3030018_

Round 1

Reviewer 1 Report

The authors of the paper entitled " Prototyping and Evaluation of Graphene-Based Piezoresistive Sensors " present interesting experimental results even if the way of preparing the samples is based on methods known in the literature. A morphological study for prepared samples (AFM or SEM, EDS analysis) would validate the quality of the samples used to make pressure sensors. The morphological study can improve a lot your paper. In introduction we must motivate this study by listing the elements of novelty in relation to literature.

Figure 1(b) – probably readers are not interested in the work table project. Here you have to give up cartoons and much more useful is the presentation of samples more realistically.

For Figure 2 authors must decide if it is J-V or I-V curves. If there are J-V curves as it appears in the text then the unit of measurement for the current density J is Current(I)/Area, or change in text from J-V to I-V.

The organization of the figures horizontally is beneficial for a plus of quality.

It is recommended a comparative study (Table) with results of other authors versus the result from these investigations to highlight the performance elements of this study given that the preparation methods are already known and no morphological evidence for samples.

It is very difficult for readers to find novelty elements as long as we do not have a comparative study to support the conclusions

Reviewer 2 Report

Dear Authors,

“Prototyping and Evaluation of Graphene-Based Piezoresistive Sensors” follows the MDPI's template.

It seems the “Featured Application: Authors are encouraged to provide a concise description of the specific application or a potential application of the work. This section is not mandatory.” Should be removed.

The introduction is a bit confusing, difficult to follow the idea. And sometimes it's not logical in my opinion, even though it contains useful technical information. But something is not connected.

In my opinion to make it more consistent some more recent, last 3 years papers, as reference, would be great.

Line 32-34: “high mechanical strength (strength and flexibility), high hardness and Young's modulus” does not make sense in my opinion and I recommend to rephrase it.

Line 37-40: It is hard to understand and follow the idea. In my opinion should be rephrased clearer.

Line 116: this is an example what is confusing me “inert atmosphere (Ar)”. I assume it is inert gas atmosphere provided by Argon. I recommend to the authors to make it clearer, straightforward.

The graphics Figure 2. are a small and hard to read.

Author Response

Please see the attachament

Reviewer 3 Report

The paper presents graphene sensor depending on the relation between resistance change and applied pressure loads. The paper does not present a new modification within the concept of graphene sensing or any contribution in the material perspective. It is just a prototype of the sensor. Some serious flows have to be considered as follows:

1- Why do the authors stop by 2 kPa? What is the response of sensor in range of loads out of the studied range in the paper?

2- Did you check if the sensor is reversible or irreversible? In details, if you removed the load, will the resistance return back to the initial result? In addition, how do your sensor reliable for repeating the loads over time?

3- Standard deviations should be presented by +/-. In addition, a comparison with other similar graphene sensor should be presented with the proper citations.

4- One microscope image of the surface of synthesized graphene film is recommended to show the uniform deposition of graphene.

5- Stress-strain curves should be presented.

6- There are some grammatic/spelling/editing mistakes, such as J(V) which has to be J-V, slashes by mistakes,...etc

Round 2

Reviewer 1 Report

The authors present an improved version of the paper "Prototyping and evaluation of graphene-based piezoresistive sensors". The comments on the first version have been corrected and, in general, this version is an important progress. Unfortunately, the authors did not give due importance to the writing of this version and a few mistakes can be reported that will require the journal team to be eliminated. Section 2 does not comply with the standard format for subsections. Figure 1 seems to be doubled and the other figures are wrongly numbered. Some figures can be resized for a more professional look. Normally this situation would involve a minor revision. The process being only one of more careful drafting I believe that from the perspective of the content of the paper can be published in this form.

Reviewer 2 Report

The authors have revised their work and improved accordingly. After performing a rigorous spell check in my opinion it should be published.

Reviewer 3 Report

All comments have been well-addressed.